Evaluation of wet and dry event’s trend and instability based on the meteorological drought index

Shaukat Muhammad Haroon 1
Al-Dousari Ahmad 2
Hussain Ijaz ijaz@qau.edu.pk 1
Faisal Muhammad 3 10
Ismail Muhammad 4
Mohamd Shoukry Alaa 5 6
Elashkar Elsayed Elsherbini 7 8
Gani Showkat 9
1 Department of Statistics, Quaid-i-Azam University , Islamabad , Pakistan
2 Department of Geography, Kuwait University , Kuwait , Kuwait
3 Faculty of Health Studies, University of Bradford , Bradford , United Kingdom
4 Department of Statistics, COMSATS University Islamabad, Lahore Campus , Lahore , Pakistan
5 Arriyadh Community College, King Saud University , Riyadh , Saudi Arabia
6 KSA workers University , Nsar , Egypt
7 Administrative Sciences Department, Community College, King Saud University , Riyadh , Saudi Arabia
8 Applied Statistics Department, Faculty of Commerce, Mansoura University , Mansoura , Egypt
9 College of Business Administration, King Saud University , Muzahimiyah , Saudi Arabia
10 Bradford Institute for Health Research, Bradford Teaching Hospitals NHS Foundation Trust , Bradford , UK
Huang Gang
Electronic publication date: 2020 Aug 24
Publication date: 2020
Volume: 8
Electronic Location ID: e9729
Received 2020 Apr 27; Accepted 2020 Jul 24
Copyright: ©2020 Shaukat et al.
Copyright year: 2020
Copyright holder: Shaukat et al.
License: This is an open access article distributed under the terms of the Creative Commons Attribution License, which permits unrestricted use, distribution, reproduction and adaptation in any medium and for any purpose provided that it is properly attributed. For attribution, the original author(s), title, publication source (PeerJ) and either DOI or URL of the article must be cited.
License URL: https://creativecommons.org/licenses/by/4.0/

Keywords: Drought, Ensemble empirical mode decomposition, Hilbert hung transformation, Multi-resolution analysis, Standardized precipitation index, Wavelet transform

Funding: Deanship of Scientific Research at King Saud University RG-1439-015 The Deanship of Scientific Research at King Saud University funded this work through research group no. RG-1439-015. The funders had no role in study design, data collection and analysis, decision to publish, or preparation of the manuscript.

==============================
A temporal imbalance in the water availability, which is consistently below average or more than average rainfall, can lead to extremely dry or wet conditions. This impacts on agricultural yields, water resources and human activities. Weather instabilities and trends of wet/dry events have not yet been explored in Pakistan. In this study, we have two-fold objectives: (1) evaluate the weather instabilities, and (2) the trend of dry/wet events of selected stations of Pakistan. To observe weather instabilities, we used Mean Marginal Hilbert Spectrum (MMHS) and Continuous Wavelet Power Spectrum (CWPS) as meteorological series are mostly non-linear and non-stationary. We used Ensemble Empirical Mode Decomposition (EEMD) for the analysis of temporal characteristics of dry/wet events. We found that all stations are facing severe weather instabilities during the short period of 5 and 10 months using MMHS method and CWPS has shown the weather instabilities during 4 to 32 months of periodicity for all stations. Ultimately, the achieved short-term weather instabilities indicated by MMHS is consistent with CWPS. In summary, these findings might be useful for water resource management and policymakers.

Introduction

In recent decades, the climate changes tend to increase the occurrence and intensity of natural perils such as heavy rainfall, flood, forest fires and droughts that indicate an increasing trend (Estrela & Vargas, 2012; Kreibich et al., 2017; Masud, Qian & Faramarzi, 2020). The consequent trend shows that precipitation makes wet places wetter and dry places drier (Trenberth, 2011), which leads to extremely dry and wet conditions that undesirably influence agricultural productions, water resources and human activities. Among these destructive climate events, drought is a natural hazard, which is a result of a prolonged shortage of precipitation, high temperature and change in the weather pattern. The substantial variability of precipitation in both space and time leads to drought. It is a challenging task to monitor and identify drought due to its dynamic spatio-temporal patterns (Shi et al., 2013).

Several drought indexes are commonly used for the detection and characterization of drought, i.e., Standardized Precipitation Index (SPI) (Mckee, Doesken & Kleist, 1993), Standardized Anomaly Index (SAI) (Katz & Glantz, 1986), Standardized Precipitation Evapotranspiration Index (SPEI), (Vicente-Serrano, Beguería & López-Moreno, 2010) and Reconnaissance Drought Index (RDI) (Tsakiris & Vangelis, 2005) etc. Mostly, SPI has been used to identify and monitor drought in previous studies (Bai et al., 2020; Caloiero & Veltri, 2019). SPI is used to measure the precipitation shortage from the long-term historical record of precipitation and represents the quantitative definition of droughts on multiple time scales i.e., 3, 6, 9, 12, 24 and 48 months. Among these time scales, 3-months SPI (SPI-3) is used for a short term, 6-months SPI (SPI-6) is used for medium-term and 48-months SPI (SPI-48) is used for long term drought analysis (Wu et al., 2007). Mostly meteorological and hydrological series have non-linear and non-stationary characteristics (Di, Yang & Wang, 2014). The Hilbert Huang Transformation (HHT) procedure is very efficient to deal with non-stationary and non-linear data and explain hydrological variability in terms of climate change (Massei & Fournier, 2012). Empirical Ensemble Mode Decomposition (EEMD) and Continuous Wavelet Transform (CWT) were applied on the two series e.g., daily Seine river flow (1950–2008) in the northern half of France and North Atlantic Oscillation (NAO) (1865–2008) to evaluate the natural variability. The coordination of EEMD with the HHT technique was found to be efficient for detecting changes in natural variation and characteristic scale of variability. The performance results of both series were indicating a similar characteristic of the scale of variability and seasonality that the amplitude is increasing since the 1970s in both series (Massei & Fournier, 2012). Bin et al. (2013) utilize the data of daily precipitation (1960–2015) of 35 meteorological stations of the lancing river basin in China. The meteorological stations were further distributed into sub-regions by using PCA and K-means clustering. Their study aimed to explore the spatial and temporal pattern with multiple time scales of meteorological drought by using the SPI. EMD and CWT were used for the analysis of temporal variability. It was found that 60% of meteorological drought variation is associated with intra-fluctuation decadal, except Chengdu station. Wang et al. (2015) utilized the precipitation data (1957–2012) of 20 meteorological stations of the North-East China Transect (NECT) region in South China. They applied the cluster analysis on SPI which is based on Discrete Wavelet Transformation (DWT) to consider the temporal evaluation of drought characteristics. It was reported that cluster analysis is efficient for spatial and temporal drought analysis and does not provide information about drought variability but provides beneficial information to water resources management and for agricultural planning. Yang et al. (2019) utilized the Modified Soil Water Deficit Index (MSWDI) of the Songnen Plain in China. Their study aimed to investigate the agricultural drought frequency and trend by HHT. Several studies suggested that HHT approach found to be more effective for decomposing the series into several components through EEMD as compared to Wavelet Transform (WT) (Massei & Fournier, 2012; Yang et al., 2019). The Mann-Kendall (MK) trend test and Spearman’s Rho (SR) test were used for the trend analysis of agricultural droughts. As a result, residuals from HHT showed the increasing trend for the year 1981 and a decreasing trend for the year 1961 to 1980. Shahid (2010) utilized the rainfall data (1958–2007) of 17 meteorological stations in Bangladesh. Two non-parametric tests, i.e., MK and Sen.’s slop (SS) tests were used on the SPI to detect a significance and magnitude in wet and dry events in Bangladesh. He found a significant decrease of dry months in monsoon and pre-monsoon.

According to a global vulnerability index report, Pakistan is in the list of the top 10 countries that are highly affected by climate change (Ullah, 2016) as its economy is heavily dependent on the agriculture sector which can severely be affected by climate changes (Ali et al., 2017). In this study, we aimed to evaluate the weather instabilities and the trend of dry/wet events of selected stations of Pakistan.

Methods and Materials

Study area

The study areas include four meteorological stations (Multan, Bahawalpur, Barkhan and Khanpur) of Pakistan. These selected stations are located in the Southern part of Pakistan, mostly very hot and mildly cold areas. Since our objective is to analyze the trend and weather instabilities with respect to time, therefore, selecting four (stations) time-series data is enough, increasing the number of stations may cause a problem in time series analysis. Monthly quantitative data of precipitation from January 1990 to December 2018 has been obtained from the Karachi Data Processing Center through the Pakistan Meteorological Department, Karachi. The geographical presentation of selected stations is displayed in Fig. 1.

Figure 1 Geographical presentation of selected stations of Southern Pakistan.

Standardized Precipitation Index at 3-months (SPI-3)

Mckee, Doesken & Kleist (1993) proposed a Standardized Precipitation Index (SPI) for defining and monitoring wet and dry events i.e., beginning, ending and intensity. The SPI is used to measure the precipitation shortage from the long-term historical record of precipitation. In this study, the cumulative precipitation series such as 3 months’ time scale is used to calculate SPI. Because, SPI-3 with a short time scale describes droughts, which is important for the agriculture sector (Caloiero, 2018; Zhang et al., 2012; Ali et al., 2017).

Method for Dry/Wet events trend analysis

Empirical Ensemble Mode Decomposition (EEMD)

Ensemble Empirical Mode Decomposition (EEMD) is an extension of Empirical Mode Decomposition (EMD) (Huang et al., 1998). EEMD was proposed by Wu & Huang (2009) and which is used to handle the mode mixing problem. Extension in EMD procedure is done through adding white noise in the time series. EEMD is a very effective procedure that minimizes the presence of mode mixing problems in the decomposition phase and is very helpful in separating frequency scales. EEMD uses the time series to decompose into several stationary Intrinsic Mode Function (IMF) and a residual component. According to Wu & Huang (2009), the EEMD procedure is as follows:

• Set the ensemble number E for each case and select the amplitude of white noise.

• Construction of white noise series ηi(t) based on the amplitude of white noise and add the white noise series ηi(t) into the time series y(t) to create a new time series yi(t), i.e., (1) yit=yt+ηit.

After this, use the traditional EMD method to decompose the new time series the IMFs. The detailed methodology of decomposing series into the IMFs by EMD is described in the Huang et al. (1998).

Methods for weather instabilities

Continuous Wavelet Power Spectrum (CWPS)

Continuous Wavelet Transformation (CWT) is used to represent the frequency in the time domain and determine the temporal variability of the series. Following (Labat, 2005), the mathematical form of CWT is as follows:

(2) Wψyγ,τ=1γ∫−∞+∞ytψ∗t−τγdt

where y(t) represents the time series, t is time and ψ∗ represent the complex conjugate of the mother wavelet function. The scale factor y is also known as the frequency that is related to the location of wavelets in the frequency domain and it is used for controlling the width of the wavelet function. Whereas, τ the parameter is related to the location of the wavelet in the time domain and it is used for the adjustment of wavelet location. Initially, Morlet wavelet is presented by Grossmann & Morlet, (2009) and its mathematical form is defined as: (3) ψt=π−14e−iγ0te−−t22

where γ0 is a non-dimensional frequency and i is the complex values. (Farge, 1992) suggested that the term γ0 is the non-dimensional frequency (angular frequency) is set 6 to satisfy the admissible condition.

Spectrum construction in the wavelet method is like the Fourier method. But it analyzes spectrum time, frequency and decomposition rather than the time and scale (Lim & Lye, 2002). According to (Torrence & Compo, 1998), the Continuous Wavelet Power Spectrum (CWPS) with scale factor is as follows: (4) Pwγ=Wψyγ,τ2.

Mean Marginal Hilbert Spectrum (MMHS)

Hilbert Hung Transformation (HHT) was offered by Huang, Long & Shen (1996). HHT is a very powerful tool and used for dealing with stationary and non-linear series. IMFs are obtained through a decomposition procedure and there is no difficulty in applying the Hilbert Transformation (HT) to obtain instantaneous frequency (IF) and instantaneous amplitude (IA). HT is only applicable for IMFs and not applicable for residual which is a monotonic function. Following (Huang et al., 2003), we can compute the Hilbert transform z(t) of time series y(t) as: (5) zt=1πP∫yt′t−t′dt′

where, P is the Cauchy value and HT be present for all the functions of LP class (Bailey & Titchmarsh, 1938). By definition, the analytical signal R(t) consists of the complex conjugate of and y(t) and z(t) (6) Rt=yt+izt=ateiφt

where, (7) at=y2t+z2t

(8) φt= arctanztyt

where, a(t) represents the superlative local fit of amplitude φ(t) and phase varying trigonometric function to y(t) (Huang et al., 2003). IF is obtained by taking the first order derivative of phase t to time t. i.e., (9) w=dθtdt.

The IF is a mono-component function in which only one component represents only one frequency. After completion of HT for each IMF, the following structure of the time series is shown as real part: (10) yt=RP∑j=1najtei∫wjtdt

HT is not applicable for the residual term (Non-IMF), because residual is a constant term. With an explanation of the HS, it is also very important to define the Mean Marginal Hilbert Spectrum (MMHS). i.e., (11) hω= ∫0THω,tdt.

The MMHS presents the contribution of total amplitude which is corresponding to each frequency value.

Non-parametric trend tests

It is very essential to extract the underlying pattern of hydrological and meteorological time series. There are several techniques for identifying patterns and analyzing hydrological meteorological time series trends. The assumptions of parametric test i.e., normality, independence and linearity are not met in the hydrological or meteorological time series. Some non-parametric techniques for dealing with non-normal and non-linear series are available in the literature, such as Mann-Kendall (MK) and Spearman’s Rho (SR) trend test.

Mann-Kendall (MK) trend test

Mann (1945) proposed a non-parametric test which is used to assess the randomness against the trend in hydrology and climatology. According to MK trend test, the null hypothesis H0 state that the (y1,y2, …, yn), are independent and identically distributed sample (there is no trend) and the alternative hypothesis H1 states that the distribution of yi and yj are not identical (there is a trend) for all i ≠ j. As a resultant, the positive value of the test statistics is an indication of an increasing trend which is increasing with respect to time. Whereas, the negative value of the test statistics is an indication of a decreasing trend which is decreasing with respect to time. Therefore, the test statistic and their theoretical calculation of MK trend test are described in Hirsch, Slack & Smith (1982).

Spearman’s Rho (SR) trend test

Spearman’s Rho (SR) trend test is based on the rank of observations. SR trend test is used to identify the absence of a linear and non-linear trend. Following (Yang et al., 2019), standardized test statistics are as follows: (12) d=1−6∑i=1nRyi−i2nn2−1

(13) zd=dn−21−d2.

Where, Ryyi is the rank of ith observation in the time series. As a resultantly negative value zd is an indication of a decreasing trend and the positive value of ith is an indication of an increasing trend. At the 5% of the level of significance (a), the null hypothesis is rejected (trend exist in the series) if zd>tn−2,1−a∕2.

Results

In this paper, the precipitation time series of four meteorological stations (Multan, Bahawalpur, Barkhan and Khanpur) have been used for explaining the wet and dry events periodicity, variability and transient trend. Initially, three months of aggregated precipitation (P-3) series are used for the computation of SPI-3. In the estimation stage, the suitable probability distribution is selected for all series of four meteorological stations. Hence, the detail of fitting suitable probability distributions on P-3 is described in Table 1.

Table 1 Several probability distributions are fitted on the P-3 series of four meteorological stations.

The CV represents the critical value of the KS text and the AD test.

Station	Distribution	Method	Parameter	BIC	KS test/CV	AD test/CV	
Multan	Exponential	MOM	a = 0.0164	3542.00	0.0313/0.0730	0.4363/2.5018	
Bahawalpur	Burr (4P)	MLE	c = 994.69, a = 1.0053, b = 52811.0, r = 0.60942	3488.32	0.0324/0.0730	0.5087/2.5018	
Barkhan	Fitgue Life (3P)	MLE	a = 0.91399, b = 82.187, r =  − 4.0011	3936.46	0.0308/0.0730	0.4367/2.5018	
Khanpur	Log-normal	MOM	r = 3.3099, b = 1.0996	3350.00	0.0357/0.0730	0.5202/2.5018	
Notes.

* The b, r and k are the scale, location and rate parameter respectively. As well as both a and c are the shape parameter.

Trend analysis of wet and dry events

Later, the SPI-3 is decomposed into IMFs by using the EEMD procedure. The decomposition level is based on the total number of observations. There are 348 total observations as the SPI-3 series consists of monthly observations from the year 1990 to 2018. According to Sang et al. (2016), log2 (348) provides eight levels of decomposition. The ensemble number is set to 100 and the amplitude of white noise is about 0.2 standard deviation has been added in SPI-3 for construction of IMFs by EEMD (Wu & Huang, 2009). Finally, by applying the EEMD method, SPI-3 of all stations is decomposed into eight IMFs and a residual term, which is depicted in Fig. 2.

Figure 2 The IMFs and a residual term were obtained by applying the ensemble empirical mode decomposition (EEMD) on SPI-3 of Multan (A), Bahawalpur (B), Barkhan (C) and Khanpur (D).

First, SPI-3 series is depicted then the eight IMFs and a residual term is depicted. The temporal variability of IMFs are depicted on the basis of this sequence for all stations.

Figure 2 shows the IMF-1 and IMF-2 intra-annual fluctuations with the duration (<1) year for all stations, though the IMF-3 and IMF-4 specifies the Intra-decadal fluctuation with periodicity between (1–3) years and (1–6) years in Multan and Khanpur, respectively. Similarly, the IMF-3, IMF-4 and IMF-5 show the intra-decadal fluctuations with the period (1–6) years in Bahawalpur and (1–4) years in Barkhan. The remaining IMFs of all stations signifies the inter-decadal fluctuations with periodicity (>10) years. These fluctuations present the instabilities of wet and dry events corresponding to each IMF. The residuals show a decreasing trend of normal dry from the year 1990 to 1998 and suddenly indicates an increasing trend of normal dry from the year 2010 to 2018 in Fig. 2A. Figure 2B shows the residual stability of a normal wet trend before the year 1998. Figure 2C shows the trend of normal wet is increasing from the year 2007. Figure 2D shows the trend of normal wet is decreasing from the year 1990 to 2007, but the trend is gradually started increasing from 2015 to onwards.

The EEMD procedures decomposed the SPI-3 on the sequence from the largest periodic component to the smallest periodic component. The first component contains the largest fluctuations of wet and dry events and gradually decreases in the fluctuations of wet and dry events on the last component. Besides, the last component (residual) indicates long term weather variability.

The significance of IMFs is tested by correlation approach (Peng, Tse & Chu, 2005). Initially, the correlation is computed between the IMFs and SPI-3 for all stations, then the threshold is applied. The correlation approach indicates that all IMFs by EEMD are statistically significant. The reconstruction of SPI-3 is based on the aggregate of all IMFs and a residual. All IMFs are utilized for reconstruction of SPI-3. Therefore, the MSE is computed between SPI-3 and the reconstructed SPI-3. It is found that the MSE of reconstructed SPI-3 has been found in Multan, Bahawalpur, Barkhan and Khanpur is 0.0024, 0.0052, 0.0034 and 0.0020, respectively.

The MK and SR trend tests are applied on SPI-3 of all stations to verify the reliability of residual trends. Initially, SPI-3 is divided into eight periods. So that the detailed information is obtained about the increasing and decreasing trends. Then the most commonly used 5% significance level is used to identify the trend. Hence, the statistical trend results of MK and SR is described in Table 2.

Table 2 Mann Kendall (MK) and Spearman rho (SR) trend test are applied on eight periods of SPI-3 of four stations.

		Period	
Station	Test	1990s	1994s	1998s	2002s	2006s	2010s	2014s	2018s	
Multan	MK	2.3370	−2.2629	2.2629	1.5772	−2.5372	0.6172	1.7143	−1.4400	
	SR	0.6947	−0.6713	0.7552	0.5315	−0.7483	0.2098	0.6154	−0.4545	
Bahawalpur	MK	1.0997	−1.7143	3.7715	−0.6172	−2.5492	1.4400	−2.1257	−1.3029	
	SR	0.5509	−0.5594	0.9510	−0.0629	−0.7940	0.4895	−0.6923	−0.4196	
Barkhan	MK	2.1995	−0.8914	2.2629	2.1257	−2.8115	2.8115	2.9486	2.9486	
	SR	0.6807	−0.3357	0.6783	0.5734	−0.7762	0.8252	0.8462	0.8112	
Khanpur	MK	0.8248	−0.7543	1.8515	−1.8515	−1.1657	3.4972	0.6172	1.9886	
	SR	0.1754	−0.3357	−0.5874	−0.5385	−0.3776	0.9231	0.2797	0.6294	
Notes.

* Two types of characters are used for identification of trend. The bold character indicates the significance of trend from MK test, while italic and bold characters indicate the significance of trend from SR test.

It is shown in Table 2 that the MK and SR indicate similar results of trend identification in all periods of Barkhan station. Whereas, MK and SR do not indicate similarity of trend identification in 2014 for Multan, in 1994 for Bahawalpur and 1998 period for Khanpur at the 5% level of significance. The positive sign is an indication of the increasing trend, while the negative sign is an indication of the decreasing trend. Moreover, the trend identification of different periods by MK and SR test is compared with the residuals of EEMD (Fig. 2) for all stations. It turns out that the SR test indicates the consistency with residuals of EEMD in mostly periods. For instance, the SR test verifies that the residuals of Multan indicate an increasing trend in 2014 while the trend does not verify by the MK test (Fig. 2A). Likewise, the residuals of Khanpur indicates a decreasing trend in 1998 which is verified by the SR test, while not verified by the MK test (Fig. 2D). SR and MK provide similar trend identification of periods for all stations except the above-mentioned stations i.e., Multan and Khanpur.

Evaluation of weather instabilities

After this, there is no difficulty to apply an HT to obtain the IF and IA. We aim to describe the detailed information of wet and dry events in the time-frequency domain corresponding to each IMF. Therefore, HT is performed on decomposed IMFs of EEMD for all stations. Generally, HT uses the IMFs to present IF and IA corresponding to each IMF. Finally, the minimum, maximum, mean and standard deviation of IA and IF are related to each IMF is presented in Table 3. Table 3 showed that the highest variation (based on the standard deviation of IF) was observed in IMF1 of all stations. It is an indication of non-stationarity of the SPI-3 for all stations.

Table 3 The HT is applied on eight IMFs of SPI-3 of all stations.

The achieved IMFs are related to EEMD. Therefore, the minimum, maximum, mean, standard deviation of IA are represented by the Min IA, Max IA, MIA and SIA respectively. Whereas, the minimum, maximum, mean and standard deviation of IF are represented by the Min IF, Max IF, MIF and SIF, respectively.

Multan	
	Instantaneous Amplitude	Instantaneous Frequency	
IMF	Min IA	Max IA	MIA	SIA	Min IF	Max IF	MIF	SIF	
IMF 1	0.018	1.252	0.359	0.192	0.000	0.493	0.243	0.141	
IMF2	0.068	1.699	0.590	0.331	0.003	0.329	0.108	0.052	
IMF 3	0.010	1.671	0.685	0.335	0.001	0.300	0.059	0.039	
IMF 4	0.159	0.710	0.419	0.132	0.000	0.120	0.031	0.011	
IMF 5	0.009	0.291	0.142	0.073	0.000	0.094	0.013	0.009	
IMF 6	0.012	0.120	0.061	0.020	0.000	0.027	0.007	0.003	
IMF 7	0.008	0.022	0.017	0.004	0.000	0.008	0.003	0.001	
IMF 8	0.000	0.010	0.007	0.002	0.001	0.029	0.001	0.003	
Bahawalpur	
IMF 1	0.020	1.234	0.335	0.201	0.001	0.486	0.226	0.142	
IMF2	0.022	1.201	0.502	0.244	0.008	0.336	0.111	0.047	
IMF 3	0.015	1.075	0.556	0.255	0.008	0.336	0.111	0.047	
IMF 4	0.168	0.633	0.406	0.135	0.000	0.288	0.056	0.034	
IMF 5	0.037	0.420	0.172	0.119	0.025	0.064	0.034	0.006	
IMF 6	0.012	0.176	0.107	0.050	0.000	0.025	0.010	0.004	
IMF 7	0.019	0.027	0.023	0.002	0.000	0.040	0.006	0.005	
IMF 8	0.000	0.005	0.004	0.001	0.001	0.025	0.001	0.003	
Barkhan	
IMF 1	0.002	1.088	0.321	0.205	0.000	0.485	0.217	0.138	
IMF2	0.007	1.327	0.411	0.265	0.001	0.339	0.109	0.050	
IMF 3	0.047	1.852	0.826	0.348	0.000	0.239	0.041	0.025	
IMF 4	0.014	0.772	0.349	0.166	0.003	0.213	0.032	0.018	
IMF 5	0.044	0.237	0.170	0.036	0.002	0.213	0.011	0.016	
IMF 6	0.011	0.313	0.210	0.075	0.002	0.095	0.005	0.008	
IMF 7	0.004	0.011	0.008	0.002	0.002	0.007	0.003	0.002	
IMF 8	0.000	0.005	0.004	0.001	0.001	0.024	0.001	0.003	
Khanpur	
IMF 1	0.009	1.025	0.371	0.190	0.002	0.487	0.246	0.148	
IMF2	0.033	1.448	0.545	0.260	0.000	0.303	0.122	0.050	
IMF 3	0.091	2.347	0.756	0.487	0.000	0.136	0.059	0.021	
IMF 4	0.074	2.009	0.515	0.424	0.000	0.088	0.029	0.011	
IMF 5	0.010	0.518	0.280	0.149	0.007	0.086	0.014	0.009	
IMF 6	0.009	0.146	0.066	0.046	0.000	0.046	0.007	0.005	
IMF 7	0.024	0.034	0.028	0.003	0.001	0.005	0.003	0.001	
IMF 8	0.000	0.006	0.004	0.001	0.001	0.219	0.003	0.017	

The MMHS provides an aggregated amount of IA in the time-frequency domain. So MMHS has been applied on the IA and IF of IMFs for all stations. The IA and IF are related to EEMD because this was achieved after HT is applied on the decomposed IMFs of SPI-3. The MMHS plots of four stations are shown in Fig. 3.

In Figs. 3A, 3B and 3D indicate the low IF of wet and dry events at 0.1 cycles per months (about log2(348) months) that is corresponding to mean IF of IMF-2 (see Table 3), while the IFs away from 0.1 are indicating comparatively low IA. Figure 3C shows the low IF of wet and dry events at 0.2 cycles per month (about <1 months) that is corresponding to the mean IF of IMF-1 (see Table 3). Besides, the IFs away from 0.2 indicate comparatively low IA. It is described that Barkhan station underwent several seasonal instabilities over a short period of 5 months. Whereas, Multan, Bahawalpur and Khanpur stations underwent weather instabilities over a short period of 10 months.

The CWPS is performed on SPI-3 of all stations for the analysis of spectral features at the different time scales. The spectral features of SPI-3 can be seen in Fig. 4.

Figure 3 MMHS plots have been shown in the analysis of weather instabilities of Multan (A), Bahawalpur (B), Barkhan (C) and Khanpur (D).

The horizontal axis specifies the IF of wet and dry events, while the vertical axis specifies the IA of wet and dry events.

Figure 4 CWPS is applied on SPI-3 for Multan (A), Bahawalpur (B), Barkhan (C) and Khanpur (D).

The colors of the wavelet power level reveal different levels of energy, and the black line indicates the 5% level of significance.

Figure 4 shows the high energy wet and dry events with a periodicity of 32 months are identified during the years (1991–2017) for all stations. Figures 4A and 4B show a spectrum of similar features of wet and dry events with a period of 16 months in Multan and Bahawalpur, respectively. During the years (1991–2010), the high energy wet and dry events are identified with a period of 16 months in Khanpur (Fig. 4D). While the high energy wet and dry events with the periodicity of 16 months are identified during the years (2002–2011) in Barkhan station (Fig. 4C). Besides, some small wavelet spectrums are indicating high energy wet and dry events with 4 to 10 months of periodicity for all stations. But the high strength of wet and dry events is found in Multan, Bahawalpur, Barkhan and Khanpur are 1.49, 1.03, 1.42 and 1.18, respectively. All these high energy wetness and dryness has been identified at the 5% level of significance.

Discussion

In recent decades, the analysis of spatial and temporal fluctuation of wet and dry events is gain more attention (Huang et al., 2014). Pakistan is an agricultural country where 70% of the people’s business depends on agriculture (Ahmad et al., 2004). In this paper, SPI-3 was selected to describe the Spatio-temporal fluctuation of wet and dry events.

Mostly meteorological and hydrological series have non-linear and non-stationary characteristics (Di, Yang & Wang, 2014). The SR and MK non-parametric tests were applied on different periods of SPI-3 for the trend analysis. It is found that the SR test is indicating a trend for most of the periods as compared to the MK test (Table 2). In recent years, inter-decadal time scales have been carefully considered about the risk of wet and dry events trend (Li et al., 2015). Besides, EEMD was applied on SPI-3 to decompose into the eight IMFs and a residual term for the trend analysis. The achieved trend of wet and dry events has been compared with SR and MK identification results. It is observed that the SR test indicates consistent results with EEMD residuals for all stations (Fig. 2 and Table 2). Moreover, the achieved IMFs have been used for reconstruction of SPI-3. As a result, the MSE of reconstructed SPI-3 is found to be less. It means that the decompose components contain the all characteristics of SPI-3. Therefore, the smaller MSE value is an indication of efficient decomposition process (Gaci, 2016).

EEMD is a purely empirical procedure and gives complete information of the temporal attributes of agricultural drought. Any non-linear and non-stationary series can be adaptively decomposed into IMF components via EEMD method, without the need to a priori basis as are Fourier and wavelet-based methods. The EEMD basis functions are known as IMF (Gaci, 2016; Ayenu-Prah & Attoh-Okine, 2009).

The standard deviation of IF shows high variability of wet and dry events in IMF-1 (Table 3). This variability is indicating non-stationarity of SPI-3 for all stations. HHT gives detailed information about IA and IF in dissimilar time scales, which is beneficial for discovering temporal variability of the wet and dry events in multiple time scales, but the wavelet analysis cannot give details. Intra wave frequency is a traditional feature of the nonlinear process which represents IF changes in one oscillation cycle. Fourier analysis cannot show the intra wave frequency, because Fourier analysis is a linear structure that applies to the non-linear process. HHT is a very effective way of describing IF in which intra wave frequency is revealed (Huang, 2014). MMHS indicates that during the short period of 5 months, the Barkhan station has experienced severe weather instabilities, while Multan, Khanpur and Bahawalpur stations have suffered severe weather instability during the 10 months (Fig. 3). The different spectrum’s of CWPS specifies 4 to 32 months of the periodicity of wetness and dryness during the years (1991–2017) for all stations. CWPS is indicating high energy weather instabilities in Multan, Barkhan, Bahawalpur and Khanpur respectively (Fig. 4). It is observed that the achieved short-term weather instabilities by MMHS provides consistent results with CWPS for all stations. By combining the results of both procedures i.e., MMHS and CWPS, it is concluded that the weather instabilities found in Multan station (1.48) is higher than Khanpur (1.18) station and in Bahawalpur (1.03) it shows least weather instabilities during the short period of 10 months. The intensity of weather instabilities for Barkhan station (1.42) during the short period of 5 months is higher as compared to other stations. It is very important to note that the MMHS provides information about weather instabilities during the short period of a month, but it does not provide information about the years in which the instabilities occurred. Whereas CWPS provides information about the monthly periodicity of weather instabilities for the time domain. The above-mentioned points illustrate the difference between MMHS and CWPS.

Conclusion

In this study, we have presented a thorough analysis of four meteorological stations (Multan, Bahawalpur, Barkhan and Khanpur) of Pakistan. We found that all stations are facing severe weather instabilities during the short period of 5 and 10 months using MMHS method.Moreover, the CWPS has shown the weather instabilities during 4 to 32 months of periodicity for all stations. Ultimately, the achieved short-term weather instabilities indicated by MMHS is consistent with CWPS. We also found that the performance of the SR test is better than the MK test for EEMD residuals trend analysis. In summary, these findings might be useful for water resource management and policymakers.

Supplemental Information

Supplemental Information 1 Codes to perform analysis

Click here for additional data file.

Supplemental Information 2 Dataset

Click here for additional data file.

Additional Information and Declarations

Competing Interests

Author Contributions

Data Availability

The authors declare there are no competing interests.

Muhammad Haroon Shaukat, Ahmad Al-Dousari and conceived and designed the experiments, performed the experiments, analyzed the data, prepared figures and/or tables, authored or reviewed drafts of the paper, and approved the final draft.

Ahmad Al-Dousari and Showkat Gani conceived and designed the experiments, authored or reviewed drafts of the paper, and approved the final draft.

Ijaz Hussain conceived and designed the experiments, performed the experiments, analyzed the data, prepared figures and/or tables, and approved the final draft.

Muhammad Faisal performed the experiments, authored or reviewed drafts of the paper, and approved the final draft.

Muhammad Ismail, Alaa Mohamd Shoukry and Elsayed Elsherbini Elashkar analyzed the data, prepared figures and/or tables, and approved the final draft.

The following information was supplied regarding data availability:

The data set and codes used for analysis are available in the Supplementary Files.

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
