# Peer review of "Evaluation of wet and dry event’s trend and instability based on the meteorological drought index"

_PeerJ, doi:10.7717/peerj.9729_

## Round 0.1 · original submission · Major Revisions

The reviewers' comments on your work have now been received. The reviewers are overall positive about your work. But one reviewer raised a serious issue regarding the presentation of the work. Please answer carefully.

Reviewer 1 ·

Basic reporting

The paper is interesting and understandable, has good spots for reflection. The structure, figures and tables are clear. And the result is valuable and it will be of interest to some researchers. However, it needs major revision before being accepted.

Experimental design

In my opinion, the number of meteorological stations is not enough. Please explain why you select these four meteorological stations of Pakistan?

Validity of the findings

No Comments

Additional comments

It is an interesting piece of work, but there still exists some problems in the manuscript. The specific suggestions are shown as follow:
(1) line 18: “Mann Kendall” should be “Mann-Kendall”. Please revise this expression throughout the manuscript.
(2) The English language should be improved. For example, line 19 “It is concluded MMHS method indicated that all stations are suffering……”.
(3) line 48: The reference doesn’t conform to PeerJ standards.
(4) Some words are incorrect in this PDF version, e.g. lines 50, 55. Please check and revise throughout the manuscript.
(5) There are many unreasonable citations in the manuscript, e.g., lines 58, 64, 70.
(6) line 65: Please use the full name of “NECT region”.
(7) line 71: “Songnen plain” should be “Songnen Plain”.
(8) line 80: “indicates” should be “indicated”.
(9) line 85: Please add “and” before “its”. Or rewrite this sentence.
(10) line 90: Please explain why you choose only four stations? Or please explain why you select four meteorological stations of Pakistan? In my opinion, the number of meteorological stations is not enough.
(11) A map of the locations of the study area and the meteorological stations should be added in this manuscript.
(12) line 126: The reference should be at the end of this sentence.
(13) lines 128, 144, 160, 167, 184, 188: “Where” should be “where”.
(14) line 176: I think the reference “Yang et al, 2019” should be deleted in this sentence.
(15) line 227: The superscript of “ith” is unreasonable.
(16) In the discussion part, the limitations of this study should be added.
(17) In the conclusion, authors should supply your future work.

Reviewer 2 ·

Basic reporting

no comment

Experimental design

no comment

Validity of the findings

no comment

Additional comments

This manuscript entitled “Evaluation of wet and dry event’s instability and trend based on meteorological drought index” by Muhammad H Shaukat et al. mainly discusses the temporal variation of wet/dry events in Pakistan using SPI calculated by observed data of four stations(Multan, Bahawalpur, Barkhan and Khanpur). The results of EEMD and MRA show prominent non-linear trend which is consistent with the Mann Kendall (MK) and Spearman’s Rho (SR) tests. And the Mean Marginal Hilbert Spectrum (MMHS) and Continuous Wavelet Power Spectrum (CWPS) are applied to detect the instabilities of wet/dry events. Similar results also are captured by Hilbert Huang Transformation (HHT). Furthermore, the differences of EEMD and MRA are also presented in this study, finding that EEMD has better performance in reconstructing SPI series. And SR test is better than MK test in describing the trend of dry/wet events. The manuscript is in general well organized and scientifically interesting for the community. Therefore, I recommend publishing the manuscript after addressed the following comments.

General Comments:
1. In Section 3(Line 246-281), the authors use the residuals of EEMD and MRA to describe the trend of dry/wet condition (Figure1, Figure2). However, the long-term trend exhibited in residual of EEMD is quite different from the one from MRA. Why?
2. In Table 2, all instantaneous frequency indexes of IMF2 and IMF3 of Bahawalpur is the same, which is very unusual. I think the authors should check their results make sure the results are right.
3. Figure 4c&4d: the location of the capture does not agree with the title in the figures, and the corresponding analysis (Section 3, Line 316-320) is confusing, too.
4.Table 3: I don’t understand what exactly the period means in table 3. For instance, which period 1990s refers to? And how is the test statistic in period 1990s calculated?
5. Section3 (Line342-346): The authors try to prove whether the trend obtained by MRA residuals (Figure 2c-d) can be verified by the MK and SR trend test (table3) in Barkhan and Khanpur. However, the analysis here is not consistent with what the table and figures show. For example, the residual shows a decrease of SPI in Barkhan in 1998 (Figure 2c) while the MK and SR tests shows a increasing trend (Table 3). It’s recommended that authors check it out and make corrections or explain it.

Minor Comments:
1. Section 3, Line 334-335: “It turns out that the SR test indicates the consistency with residuals of EEMD in mostly periods as compared to MRA.” “compared to MRA” should be “compared to MK”.
2. Authors should also pay attention to the citation format. For example, in Section 1, Line 59, “(Bin Li et al., 2013) utilize the data of daily precipitation (1960-2015) of 35 meteorological stations of the lancing river basin in China” It should be changed to “Bin Li et al. (2013) utilize the data of daily precipitation (1960-2015) of 35 meteorological stations of the lancing river basin in China”.
3. Reference (Line484-486 & Line503-504): The DOI is missing.

Reviewer 3 ·

Basic reporting

no comment

Experimental design

no comment

Validity of the findings

no comment

Additional comments

Comment on the “Evaluation of wet and dry event’s instability and trend based on meteorological drought index”

In this study, the authors aim to evaluate variability and trend of the wet/dry event of the four stations in Pakistan by using several methods. The differences between the results based on different method are also contrasted. Through careful reviewing, the reviewer think that the present manuscript is badly written and lacks of clear logicality. And importantly, the authors paid more attention to the method rather than the results or the merit of this study. Therefore, I recommend the article a rejection at least a resubmission after a major revision.
Other comments:
1. In the abstract, the authors seem to evaluate the applicability of the method in detecting the variability of the wet/dry event, because more than 2/3 part is used to the introduce the methods.
2. The reviewer can not obtain clear goals of the present work, as well as the present understanding and main issues in the literature.
3. I suggest the authors focus on one or two methods and highlight the new findings.

---

## Round 0.2 · Major Revisions

Since there is a reviewer who remains unconvinced by your revision please address your reply to this reviewer item by item.

Reviewer 1 ·

Basic reporting

No Comments

Experimental design

No Comments

Validity of the findings

No Comments

Additional comments

The authors have addressed all my comments.

Reviewer 2 ·

Basic reporting

no comment

Experimental design

no comment

Validity of the findings

no comment

Additional comments

The authors modified the manuscript according to my comments/suggestions. I therefore recommend an acceptance of publication.

Reviewer 3 ·

Basic reporting

The authors revised the manuscript mainly by polishing the manuscript. However, the weak analyses and weak contrast or merit are still major shortcomings of the manuscript, which has been pointed out in the previous review cycle. The revised manuscript still does not convince me enough. Therefore, I would suggest rejecting the manuscript.

Experimental design

no comment

Validity of the findings

no comment

Additional comments

The authors revised the manuscript mainly by polishing the manuscript. However, the weak analyses and weak contrast or merit are still major shortcomings of the manuscript, which has been pointed out in the previous review cycle. The revised manuscript still does not convince me enough. Therefore, I would suggest rejecting the manuscript.

---

## Round 0.3 · accepted · Accept

After multiple reviews by 3 reviewers, I am glad that your paper has been accepted. Congratulations!

Reviewer 3 ·

Basic reporting

no comment

Experimental design

no comment

Validity of the findings

no comment

Additional comments

I have no further comments on the manuscript.